# Primer on Reproducible Research in R: Enhancing Transparency and Scientific Rigor

**Mushfiqul Anwar Siraji** [1,2,*] **and Munia Rahman** [3]

1   Department of Psychology, Jeffery Cheah School of Medicine and Health Science, Monash University Malaysia, Jalan Lagoon Selatan, Bandar Sunway, Selangor Darul Ehsan 47500, Malaysia
2   Department of History and Psychology, School of Humanities and Social Sciences, North South University, Dhaka 1229, Bangladesh
3   Department of Psychology, University of Dhaka, Dhaka 1000, Bangladesh
*   Correspondence: mushfiqul.siraji@northsouth.edu

**Abstract:** Achieving research reproducibility is a precarious aspect of scientific practice. However, many studies across disciplines fail to be fully reproduced due to inadequate dissemination methods. Traditional publication practices often fail to provide a comprehensive description of the research context and procedures, hindering reproducibility. To address these challenges, this article presents a tutorial on reproducible research using the R programming language. The tutorial aims to equip researchers, including those with limited coding knowledge, with the necessary skills to enhance reproducibility in their work. It covers three essential components: version control using Git, dynamic document creation using rmarkdown, and managing R package dependencies with renv. The tutorial also provides insights into sharing reproducible research and offers specific considerations for the field of sleep and chronobiology research. By following the tutorial, researchers can adopt practices that enhance the transparency, rigor, and replicability of their work, contributing to a culture of reproducible research and advancing scientific knowledge.

**Keywords:** reproducible research; R; version control; Git; GitHub; renv

## 1. Introduction

The reproducibility of findings is the gold standard in scientific practice [1]. However, numerous examples of studies from various disciplines cannot be fully replicated. This failure to reproduce findings can be largely attributed to the disseminating methods [2]. The common practice of disseminating findings is publishing them in journals, articles, books, conferences, or web pages. These methods are successful in accurately presenting scientific findings and convincing readers that these findings are correct, but they fail to truly describe the full research context and procedures [3]. Over the past ~50 years, it has been observed that experimental computational research is highly involved with skilled, sophisticated software packages that help researchers to partake in advanced and novel methods [3]. A full research description involves a description of these methods and the software environment, code, data and analysis steps that produced the results [4]. The conventional ways of reporting findings provide a scant description of these components, thus hindering their reproducibility. This issue was demonstrated in an investigation of replicating results from openly available materials in the journal *Cognition* [5]. Among the 35 published articles that provided open code and data, the results of 22 articles could be replicated, albeit with assistance from the original authors in 11 cases. However, for 13 articles, at least one outcome could not be reproduced, even with the aid of the original authors. Another study examining 62 registered reports revealed that only 41 had accessible data, and merely 37 had analysis scripts available [6]. Out of these scripts, only 31 could be executed successfully, and the results of only 21 articles could be reproduced within a

reasonable timeframe. These unsuccessful attempts at reproducing findings underscore the urgent need for widely accepted standards of reproducibility.

## 1.1. Reproducible Research

The replicability of scientific findings is considered as the key component, perhaps from the 1200s [7], that marks the difference between science and nonscience [8]. This allows us to scrutinize the reported findings and verify them as genuine contributions [9]. To allow the replicability of research, it should be reproducible; hence comes the concept of *Reproducible Research.* Reproducible research, coined by geophysicist Jon Claerbout in 1990 [3], is the practice of conducting scientific investigations in a reproducible way to replicate the steps and the findings independently. Reproducible research plays a crucial role in evaluating research findings and minimizing the need to duplicate efforts in data collection and procedure development. By adhering to reproducibility principles, researchers can efficiently assess and validate results without having to invest excessive time in re-gathering data or re-establishing previously determined procedures. While describing the research in a reproducible way, one must attain a balance of adequacy and efficiency. How to attain this balance is an issue of high debate [10,11]. To attain reproducibility, researchers at least make sufficient information available to allow any researcher to reproduce the result independently using the same procedure. The process of informing other researchers about the research and relevant steps to reproduce the findings should be easy to follow [12].

Focusing on reproducible research has other perks as well. Emphasizing reproducibility fosters better work habits, improves planning and organization, and promotes higher-quality data and source code [4]. It encourages error detection at an earlier stage and facilitates error identification through clear documentation. Storing the research in an accessible manner enables other researchers to access and utilize the data and source code while also facilitating revisions by avoiding duplication of personal efforts. The practice of reproducible research facilitates collaboration and knowledge sharing among colleagues. The usefulness of research, as measured by citation frequency, is significantly enhanced when it is fully reproducible [13,14].

## 1.2. Reprodicibility in Sleep and Chronobiology Research

Sleep and chronobiology research lacks reproducibility in many instances. For instance, among the 59 studies included in the systematic review of Siraji and Kalavally [15], no studies reported pre-registration; data were instantly available for 1% of the studies, and analysis codes were available for 1% of the studies. Additionally, reporting on seasonality was scarcely mentioned. Schöllhorna and Stefani [16], in their recent review, pointed out the importance of reporting seasonality while investigating the influence of light on sleep and circadian rhythms. Study results obtained during summer cannot be directly reproducible if conducted in winter, due to seasonal variations. While measuring parameters like alertness, executive functioning and decision making, studies often lack a proper description of the task executed. Computer-based tasks such as psychomotor vigilance tasks, n-back tasks, and go-no-go tasks vary significantly, and the outcomes may be susceptible to the hardware, coding system and execution platforms. So, it is important to detail the task with all the required information. An example of how to report computer-based cognitive tasks can be found in the Supplementary Materials of Grant and Kent [17]. In studies investigating the influence of light exposure on sleep, cognition and chronotypes, it is also important to describe the variables and their manipulation. Lok and Joyce [18] provide a good example of detailing study variables and their manipulation to increase the reproducibility of works.

## 1.3. Reproducible Research and R

There have been several approaches developed to increase the reproducibility of research work. These approaches include providing codes within a text document and using markup languages to produce all texts, figures, codes, and equations used in the

research [19,20]. While the approaches mentioned above are effective in achieving reproducibility, they may not be practical for non-programming experimental scientists who rely on third-party or commercial software tools [3]. These scientists often face challenges in reproducing research, due to limited access to underlying codes or dependencies. Frequently, these documents are distributed in a fragmented manner, with data and code residing in separate appendices or referenced web pages. This disjointed approach places the burden on individual users to navigate between the text, data, and code components themselves [20]. However, there are easy-to-learn ways that utilize R language [21] to develop a dynamic document that combines text, data, and codes.

R [21] boasts a thriving development community that consistently expands its capabilities. Being distributed under the GNU General Public License (GPL), R [21] is free, open-source, and accessible to all, thus making it a perfect tool for reproducibility. Using R [22], rmarkdown [22], and RStudio—an integrated development environment (IDE) for R with active integration with version control platforms—enables researchers to directly link their analyses, results and codes which are used to generate those results together, thus making it easy to trace the steps while reproducing and to help minimize the threats to reproducibility [23]. Version control helps to trace modifications and additions to, or the deletion of, codes, datasets and other materials over time and allows us to view, compare, and revert to previous versions of files or documents. RStudio offers a very smooth version control system using Git, allowing the researchers to easily keep trace of the changes made to files or documents. Rmarkdown offers an easy and intuitive way to create dynamic documents that hold texts, codes and data together. Such a coherent presentation helps to reduce errors related to the inconsistency of codes, provides a better understanding of analysis steps and provides a detailed track of all packages and dependencies used in the analysis. With all these intricacies, R can provide long-term, cross-platform reproducibility.

## 2. A Tutorial on Reproducible Research

This paper offers a user-friendly tutorial that illustrates how R can be utilized to achieve research reproducibility. The tutorial specifically targets researchers with minimal or no coding knowledge, aiming to provide them with the necessary foundation to attain reproducibility. The focus of the discussion is on essential primers that enable readers to grasp the basics. As researchers gain proficiency in utilizing reproducibility tools, they can progressively enhance the reproducibility of their work. This tutorial incorporates three basic components to achieve reproducibility: (a) version control, (b) the creation of dynamic document creation, and (c) managing R package dependencies. These three components will be discussed centered on the R environment [21]. We will focus on Git for version control [24], rmarkdown [22] for developing dynamic documents, and renv [25] for the containerization. In this tutorial, RStudio is used as the integrated development environment (IDE) for R. We will finish the tutorial with some pointers on how to share reproducible research with other researchers and provide specific considerations for the field of sleep and chronobiology research.

### 2.1. Installing the Software

Our discussion will start with the installation of the main software. R and RStudio are free, open-source and available for Windows, Mac, and Linux operating systems, thus offering the best solution for cross-platform reproducibility. R and RStudio can be downloaded from https://posit.co/download/rstudio-desktop (accessed on 14 December 2023). Before we install RStudio, R is required to be installed first. We also need to install a Tex Distribution to help us knit our dynamic document into PDF. Tex distribution can be downloaded from https://www.latex-project.org/get (accessed on 14 December 2023) Another way, and perhaps the most convenient way, to install the required Tex distribution for rmarkdown is installing TinyTex with R package *tinytex* [26]. Tinytex can be installed by running *tinytex::install_tinytex()* in the R console. For version control, we need to install

version Git, which can be downloaded and installed from https://git-scm.com/downloads (accessed on 14 December 2023).

*2.2. Version Control*

To enable easy and smooth version control, we will integrate git and RStudio via GitHub—a web-based platform that provides version control. First, we need to create a free user account in GitHub and check whether R, RStudio and Git are properly installed on our computer. Second, we need to create a repository for our research project in GitHub. This repository will contain all the relevant files and materials for our research. Figure 1 describes the steps to create a repository on GitHub.

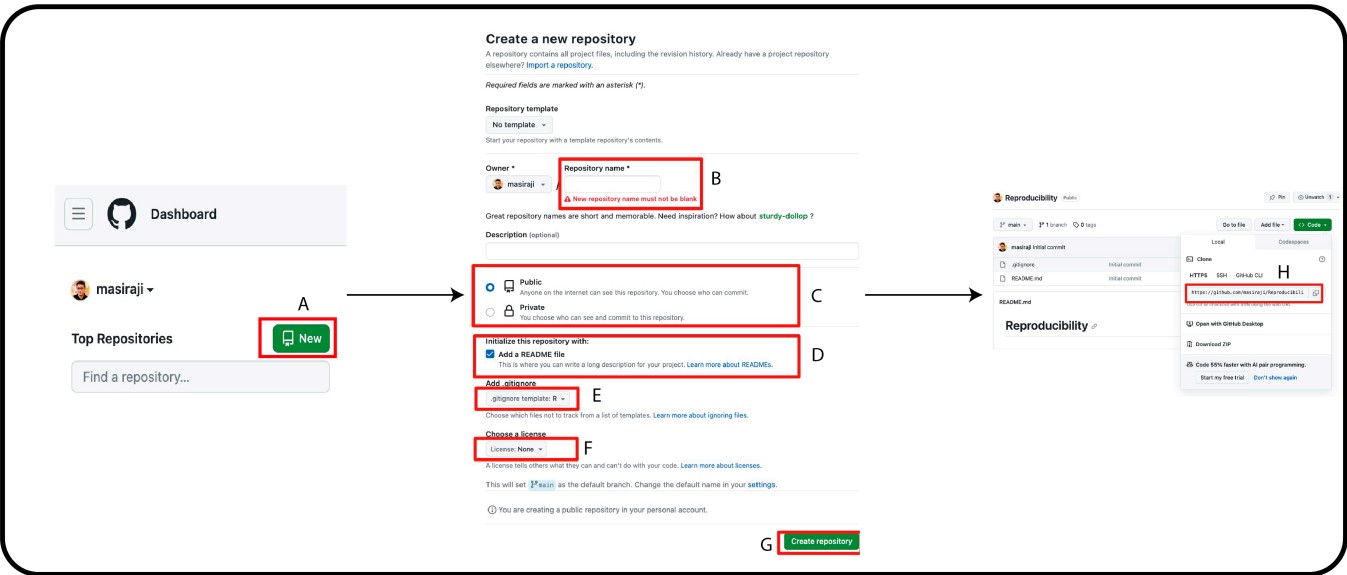

**Figure 1.** (A) On GitHub page's left column, click "New". This will lead you to a new page. (B) On this new page, write your preferred repository name. (C) You can keep the visibility of the repository as public or private. This can be changed anytime you want after the creation of the repository. (D) Creating a README file with the appropriate repository description is recommended. (E) Add an R-based gitignore file. (F) GitHub provides options to select the appropriate license for your work. (G) Once you are satisfied with all the options, click "Create repository" to create the repository. (H) Once the repository is created, GiHub will automatically lead you to the repository landing page. In the landing page, click on "Code" and copy the repository URL.

Third, after we have created the online repository, we go to RStudio > File > New Project > Version Control > Git. In "Repository URL", paste the URL of the new GitHub repository. We keep the project directory name unchanged and modify the address *in "create project as subdirectory"* to where we want to save the GitHub repository locally on our computer (Figure 2). Once the local repository is created, there will be an R project file with an "*.Rproj*" extension. This R project file is the container of our research documents, such as data files, R scripts, documents, and other materials. The working directory, environment preferences and project-specific options can all be conveniently saved in the RProject file. When the researchers want to work on a research project in their local repository, it is advised to open the RProject file in RStudio, which will keep the working environment constant and promote reproducibility.

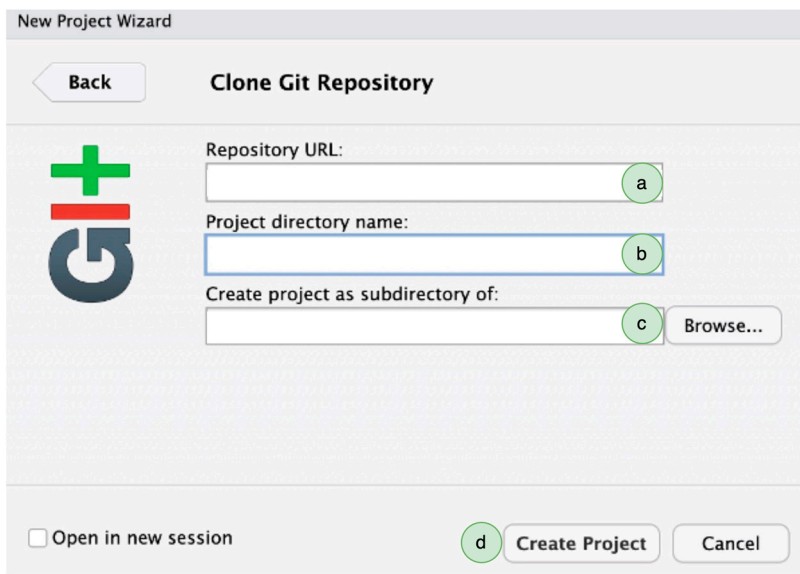

**Figure 2.** Creating a Git-enabled project and saving it on a local computer. (a) Paste the URL of the GitHub repository. (b) The name of the online repository will appear here. (c) Select the local address for the repository. (d) Click on "Create Project", and it will create a local copy of the online repository.

Once we have connected the GitHub online repository to our local computer, we can change, modify and delete our files on our local computer and push the changes to the online repository to synchronize the files. This process also allows for the simultaneous collaboration of a team of researchers. To avoid conflicts of changes being made, it is optimum to follow a specific workflow when we synchronize our local repository to the online one.

- We open our local project by opening the file that has an "*.Rproj*" extension. By doing so, our project will open in the RStudio.
- Before introducing any changes, we click on the Git tab in RStudio and click on "*pull*" to synchronize our local database to the online database.
- Once our local repository is synced, we start working on our local files, commit the changes and click "pull" to get the latest changes made in the repository (if any). Lastly, push your changes to the online repository to synchronize.

It helps reproducibility if we have an organized folder structure system in the repository that makes navigation easy in the future. Figure 3 provides an example folder organization structure. The repository will contain a R project file (file with "*.Rproj*" extension), a folder for raw data, a folder for processed data, a folder that will house all relevant data visualization and a dynamic document (file with ".rmd" extension) that will contain all our texts, analysis codes and data together. It is also advisable to have a README file that will detail every component that exists in the repository. It is good practice to provide a brief description of the purpose of the repository. An introductory line such as "This repository contains the data and analysis codes for the manuscript titled XYZ" is a working example. Providing the website links containing the additional materials of the manuscript is also recommended. Provide the description of file directories in the README file detailing the file type and how to use it to reproduce the work.

The "*renv*" folder and "*renv.lock*" file in the repository will appear after we activate the R package dependency management. We will discuss R package dependency management later in this tutorial. Also, it is advisable to add a license to the repository that will allow others to legally reproduce the work. We will discuss these aspects of sharing one's research work at the end of this tutorial. Examples of reproducible research works that have an organized repository structure can be found in the work of Siraji and Haque [27] and Siraji

and Lazar [28]. Readers can find more detailed instructions on version control using R Studio, Git and GitHub at https://happygitwithr.com (accessed on 14 December 2023).

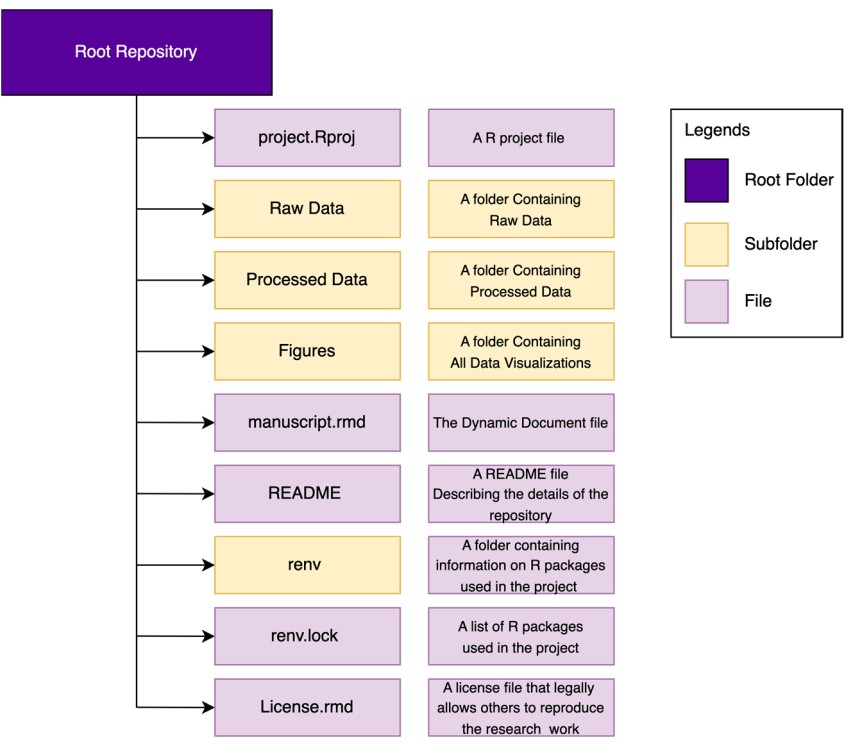

**Figure 3.** Repository structure to facilitate reproducibility.

## *2.3. Creating a Dynamic Document*

Converting computational results into a comprehensible format, such as a technical report, presentation, or manuscript, can be a laborious and error-prone process. Common mistakes arise from errors in copying the codes, inaccurate rounding, or failing to update the results in the manuscript when the underlying computer code and calculated results have been modified. To avoid these common mistakes and increase the reproducibility of the results and the reports, the literate programming paradigm suggests using a dynamic document that holds all text and codes knitted together [29]. With the development of the R programming language, we now have several advanced packages that create dynamic documents in R. *rmarkdown* [22] is such a versatile package that helps us to knit our research results and findings as documents (docx, pdf, pptx, epub, and html). Using *rmarkdown*, we can write our narratives and code in an intermingled form. Such presentation of an analysis code with narratives helps us to communicate our analysis steps to other researchers. For example, we can easily communicate what modifications to the dataset are made to address outliers and missing values and pinpoint at what stage of our analysis we introduced the modification. *rmarkdown* allows us to put in codes in two specific ways (a) using chunks (a collection of codes) and (b) inline (putting a short code within the text). The combination of these two methods helps us to attain the highest form of reproducibility. For example, we can put our entire analysis code in chunks and use an inline code for dynamic integration to embed R codes directly within texts. The inclusion of an inline code in dynamic documents enhances its reproducibility. It enables readers to view the exact R code employed to generate the presented results, ensuring transparency and facilitating the replication of analyses. It is possible to knit the dynamic document according to the preferred formatting styles of publication houses. There are several packages that extend the applicability of rmarkdown-based dynamic documents. Noteworthy among the packages available in R are *papaja* [30] and *tabledown* [31] and *stargazer* [32]. The *papaja* package offers additional functions that facilitate the formatting of documents in accordance with the American

Psychological Association (APA) style, including the creation of journal-style final typeset formats. On the other hand, the *tabledown* and *stargazer* package provides convenient tools for generating publication-ready tables and reports of statistical models. For a detailed guide on how to use rmarkdown, please read Xie and Dervieux [33],

Important things to remember while creating dynamic documents:

- Always create a chunk containing all R packages and dependencies used in the analysis and reported in the dynamic document.
- Clearly document the sources of data, including file names, URLs, or database connections. Provide details about data preprocessing steps, transformations, and any modifications applied.
- While coding, practice a reproducible coding style with relevant comments and descriptions of each code chunk. Remember to set seed values for random number generation and capture the session information, including the R version, package versions, and system details, at the beginning of the document.

### 2.4. Managing R Packages Dependencies

The R ecosystem's *renv* package [25] is an effective tool for managing project-specific R package dependencies, which improves reproducibility. It enables the creation of isolated, independent settings by researchers, ensuring consistent and repeatable outcomes across various computing configurations. Researchers can record the current state of the R packages used in a project, including their versions and dependencies, using renv. If package upgrades or modifications take place over time, this guarantees that the project can be simply replicated. *renv* assists in avoiding problems brought on by incompatible package versions, unavailable packages, or modifications to package functionality.

A step-by-step description of how to use renv:

- *renv* can be installed by running *install.packages("renv")* in the RStudio console.
- To enable renv-based package dependency, navigate towards the project folder in the local computer and open the R project file with an "*.Rporj*" extension. Once the project is open in RStudio, run *renv::init() and* initiate renv. This will create a new *renv* folder in your project directory.
- To make a record of your current status, i.e., the R version, the packages used and the corresponding version number, run *renv::snapshot()* in the console. This will take a snapshot of the current status of the R environment and create a *renv.lock* file in the project directory, which lists the packages and their specific versions used in the project.
- Once the *renv.lock* file is created, remember to update it whenever a new package is added or existing packages are updated within the project. To do this, simply run the *renv::snapshot()* command. This ensures that the *renv.lock* file accurately reflects the current state of the project's package dependencies.
- Make sure the GitHub online project repository contains the updated *renv folder* and *renv.lock* file.
- To recreate the project environment on another system or to revert to a previous state, run *renv::restore()* in the R console. This will install the packages specified in the *renv.lock file*.

### 2.5. Sharing the Reproducible Work

Once we have a structure to facilitate research reproducibility, attention should be given to how we share our work with others. In this paper, we focused on using a free GitHub online repository that can host our research materials securely. This online repository can be open as private only, visible to the researcher(s), or public/open to all. To allow others to freely use the shared materials while preserving author attribution, it is crucial to choose the right licenses for codes, data, and media. It is recommended to use the MIT license (or comparable permissive licenses) for computer codes and the Creative Commons—Attribution license (CC-BY) for texts, R Markdown files, and media [23]. These

licenses give users the most freedom while still demanding attribution and are compliant with the Reproducible Research Standard [34]. choosealicense.com is a potential resource to help determine which license could be employed. When reporting findings in journals, conferences or webpages, researchers can share the URL of their public GitHub repository link that holds all their research materials in an organized and reproducible manner. The works of Siraji and Haque [27] and Siraji and Lazar [28] are two working examples of good practices for reproducible research and how to share research works with others.

*2.6. Reproducible Research: Sleep and Chronobiology*

Thus far, we have discussed how to attain reproducibility in research work in general. However, every discipline is unique in its own nature and demands tailored procedures to attain full reproducibility. Here, we recommend some additional steps to ensure reproducibility in sleep and chronobiology research.

- Pre-registration: Pre-registration of research work involves making research goals, theories, and analysis procedures available to the public before data collection. This practice encourages openness, reduces bias, and prevents selective reporting, ultimately facilitating reproducibility. The Open Science Framework (OSF) is a collaborative research management platform that supports open and reproducible scientific practices. With its pre-registration tool, researchers can publicly document their research plans, hypotheses, and analysis protocols prior to conducting the study, ensuring transparency and minimizing biases.
- Standardized Assessment Methods: Using and developing standardized assessment methods for evaluating the parameters of sleep and chronobiology is essential. Utilizing standardized tools and methods helps with reproducibility and assures comparability between investigations.
- Providing a Detailed Study Protocol: For sleep and chronobiology research to be reproducible, it is crucial to provide complete descriptions of the research protocols. This includes thorough documentation of inclusion–exclusion criteria, the methods used to collect the data (such as actigraphy and polysomnography), the actions involved in data preprocessing and the demographics of the participants. Tir and White [35] provide a detailed guideline on how to report the demographics of the participants in sleep and chronobiology experiments. Special attention should be given if the research is involved with light exposure. For a detailed guideline on how to report on light exposure in human sleep and chronobiology experiments, please see the work of Spitschan and Stefani [36].
- Open Access to Datasets: Using open access procedures for disseminating sleep and chronobiology data encourages reproducibility. By making anonymized data accessible, other researchers are able to confirm results, carry out their own analyses, and consider new lines of inquiry. Utilizing repositories such as GitHub or the Open Science Framework (OSF) for publicly storing data promotes field-wide transparency, collaboration and reproducibility.
- Sharing Sleep Monitoring Tools and Algorithms: Researchers should disclose details regarding the hardware configurations and data processing methods of the monitoring devices they utilized in their studies to improve repeatability. Guaranteeing the consistency and reproducibility of results makes it possible for others to reproduce data collection procedures and use comparative analysis approaches.

## 3. Conclusions

The main goal of this paper was to provide a tutorial that provides a primer on the process of practicing reproducibility in research using R. We centered our tutorial on R and discussed three components, version control, developing dynamic documents and managing dependencies, which enable other researchers to easily follow all the steps taken in research, from conception to manuscript writing, thus facilitating reproducibility. However, this tutorial deals with the very basic components of R-based reproducibility and

can be thought of as the starting point for researchers who want to learn about R-based research reproducibility. We recommend reading the works of Gandrud [2], Peikert and Van Lissa [23], and Xie and Dervieux [33] to know the advanced techniques used in R to improve research reproducibility.

**Author Contributions:** Conceptualization, M.A.S. and M.R.; resources, M.A.S. and M.R; writing—original draft preparation, M.A.S. and M.R.; writing—review and editing, M.A.S.; visualization, M.A.S. and M.R.; supervision, M.A.S. All authors have read and agreed to the published version of the manuscript.

**Funding:** This research received no external funding.

**Institutional Review Board Statement:** Not applicable.

**Informed Consent Statement:** Not applicable.

**Data Availability Statement:** No new data were created or analyzed in this study. Data sharing is not applicable to this article.

**Conflicts of Interest:** The authors declare no conflict of interest.

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
