# Peer review of "Primer on Reproducible Research in R: Enhancing Transparency and Scientific Rigor"

_2624-5175, doi:10.3390/clockssleep6010001_

Round 1

Reviewer 1 Report

Comments and Suggestions for Authors

A well-written tutorial

Comments on the Quality of English Language

Abstract: Please delete "triumphing"

Author Response

Thank you so much for your time and consideration. Please see the attachment

Reviewer 2 Report

Comments and Suggestions for Authors

The authors were trying to create a user manual for researchers who want to make their papers are more reproducible. The target tools are: git as a version control system, GitHub as an IT-hosting, rmarkdown as a dynamical documentation tool, R as a main programming language, RStudio as an IDE.

The main issue of this paper is the not scientific paper is just a tutorial that reader can find on the Web. Moreover, the tutorial is very poor, more significant parts are described without details and examples.

I suggest adding more reviews of the papers in the research reproducibility field. The authors also must add more details and examples in their tutorial. Moreover, the authors need to add real-life example.

Author Response

(The authors gave the same response as above.)

Reviewer 3 Report

Comments and Suggestions for Authors

This work addresses a very interesting question: the reproducibility in research. However, the work is basically a presentation of a tutorial that provides a primer on the process of practicing reproducibility in research using R. A scientific article and a tutorial have absolutely different natures and characteristics. In this sense, it becomes necessary to consolidate the literature review related to the problem in question. It would be interesting to carry out a bibliometric study that supports the detected needs in terms of the reproducibility of investigations. This research could be done in the area of Sleep and Chronobiology.

The criteria that led to the selection of the journal Cognition to develop this study and how the analyzed articles were selected in unclear.

It is important to review the structure of the Introduction. On the one hand, it is unnecessary to include in the Introduction the information presented in lines 39-45, more generic information should be presented. On the other hand, the Introduction includes a point 1.2, without there being a point 1.1.

In summary, it is suggested that the authors deepen the theme under analysis, present a bibliometric research in the area of ​​Sleep and Chronobiology that justifies the need for the tutorial presented and that they send all the technical aspects to the appendix.

Author Response

(The authors gave the same response as above.)

Round 2

Reviewer 2 Report

Comments and Suggestions for Authors

The authors were trying to create a user manual for researchers who want to make their papers are more reproducible. The target tools are: git as a version control system, GitHub as an IT-hosting, rmarkdown as a dynamical documentation tool, R as a main programming language, RStudio as an IDE.

The main issue of this paper is the not scientific paper is just a tutorial that reader can find on the Web. Moreover, the tutorial is very poor, more significant parts are described without details and examples.

I suggest adding more details and examples in their tutorial. Moreover, the authors need to add real-life example.

Author Response

Dear Reviewer, 

Thank you for your valuable feedback. We totally agree with you on the note—this is a through-and-through tutorial, not a scientific work. We felt the necessity of writing this tutorial as a contribution to science, enabling basic researchers to enhance the reproducibility of their work.

It is true that there are lots of web tutorials available. This tutorial provides rather simplified steps to attain reproducibility. We focused on the basic steps; hence, we might call it simple rather than poor. We believe our tutorial provided very easy-to-follow steps for the non-coder researcher community to enhance their reproducibility. We have provided and mentioned resources wherever possible to provide more in-depth steps for those who want to know more. Additionally, several real-world repositories are cited to provide real-life examples.
complexity of this tutorial. We wanted it to be a primer, not a comprehensive module. Hence, we do not opt for adding more details, which may increase the

Thank you for your comments and consideration. We are grateful to you.

Regards
Authors

Reviewer 3 Report

Comments and Suggestions for Authors

This is a second review. The authors considered the suggestions received. The changes introduced allowed to improve the quality of the manuscript.

Author Response

Dear Reviewer,

Thank you for your feedback. We are happy that you think our modifications are satisfactory. 

Round 3

Reviewer 2 Report

Comments and Suggestions for Authors

If the authors want to write a truly necessary and useful guide, then they should do the following work:

1. Carry out a test-case study on the journal related topic.

2. Add explanations and images to each step of the study.

3. Use technologies to perform a test case: git, R, latex, etc.

I expect to see a very detailed guide. Authors must describe each step of the research in great detail. All actions must be described and contain examples. I think that the format of the article will allow the authors to produce a guide that will be useful to researchers outside the IT field.

Author Response

Respected Reviewer

Thank you for your comments. Yes, your recommendation would improve the relevance of the training module. Unfortunately, we don't have the resources to run a test-case. Our intention was to inform the readers with the steps to increase reproducibility. Several repositories are mentioned in this tutorial that used these steps to increase reproducibility. Those could serve as a better example than a test case.
Thank you for your time and consideration. 

Round 4

Reviewer 2 Report

Comments and Suggestions for Authors

If the authors want to write a truly necessary and useful guide, then they should do the following work:

1. Carry out a test-case study on the journal related topic.

2. Add explanations and images to each step of the study.

3. Use technologies to perform a test case: git, R, latex, etc.

I expect to see a very detailed guide. Authors must describe each step of the research in great detail. All actions must be described and contain examples. I think that the format of the article will allow the authors to produce a guide that will be useful to researchers outside the IT field.

Author Response

Respected Reviewer,

Your review comments from round 3 and round 4 are the same. As we have responded before, running a test case is out of our scope. We have added adequate references that are quite capable to guiding the readers in engaging reproducible research. We think we have added adequate illustrations as well. Any more addition to the figures would break the reader's flow. We have added enough resources on the use of Git and R. Any more details would make the paper complicated. 
We strongly think that our paper is well versed in guiding people outside the IT to reproduce their work.

Thank you for your valuable feedback.